# Dynamic Release Characteristics and Kinetics of a Persulfate Sustained-Release Material

**DOI:** 10.3390/toxics11100829

**Published:** 2023-09-30

**Authors:** Xueqiang Zhu, Hanghang Ji, Gang Hua, Lai Zhou

**Affiliations:** 1Engineering Research Center of Mine Ecological Restoration, Ministry of Education, Xuzhou 221116, China; zhuxq0615@163.com (X.Z.); ts21160139p31@cumt.edu.cn (H.J.); huag.jssy@sinopec.com (G.H.); 2School of Environment and Spatial Informatics, China University of Mining and Technology, Xuzhou 221116, China

**Keywords:** persulfate, sustained release, model fitting, soil and groundwater remediation, in situ chemical oxidation

## Abstract

Sustained-release materials are increasingly being used in the delivery of oxidants for in situ chemical oxidation (ISCO) for groundwater remediation. Successful implementation of sustained-release materials depends on a clear understanding of the mechanism and kinetics of sustained release. In this research, a columnar sustained-release material (PS@PW) was prepared with paraffin wax and sodium persulfate (PS), and column experiments were performed to investigate the impacts of the PS@PW diameter and PS/PW mass ratio on PS release. The results demonstrated that a reduction in diameter led to an increase in both the rate and proportion of PS release, as well as a diminished lifespan of release. The release process followed the second-order kinetics, and the release rate constant was positively correlated with the PS@PW diameter. A matrix boundary diffusion model was utilized to determine the PS@PW diffusion coefficient of the PS release process, and the release lifespan of a material with a length of 500 mm and a diameter of 80 mm was predicted to be more than 280 days. In general, this research provided a better understanding of the release characteristics and kinetics of persulfate from a sustained-release system and could lead to the development of columnar PS@PW as a practical oxidant for in situ chemical oxidation of contaminated aquifers.

## 1. Introduction

In situ chemical oxidation (ISCO) is a cost-effective technology able to eliminate organic contaminants from the soil and groundwater [1,2]. Permanganate, Fenton reagents, ozone and persulfate (PS) are commonly used specific oxidants in ISCO [3,4]. Compared to other oxidants, PS is more effective and reliable as an oxidant in ISCO due to its ability to be applied across a wide range of pH levels [5]. In addition, the degradation process does not produce secondary contaminations and undesirable precipitates [6,7]. PS has a standard oxidation reduction potential of 2.01 V [8,9]. Thus, PS could react with organic compounds; however, the rate of degradation of certain pollutants is slow [10,11]. Studies have shown that PS can produce sulfate free radicals (SO_4_^−^·), as indicated by Equations (1) and (2) [12,13], having a higher oxidation reduction potential of 2.6 V [14,15]. As a result, PS has become a favored oxidant for chemical oxidation to remediate contaminated groundwater, and it has been demonstrated to have the capability to degrade numerous organic pollutants [16].
(1)$$S_{2}O_{8}^{2-}\overset{activator\left(e^{-}\right)}{\to }{\mathrm{SO}}_{4}^{-}\cdot +{\mathrm{SO}}_{4}^{2-}$$
(2)$$S_{2}O_{8}^{2-}\overset{energy\;input\left(e^{-}\right)}{\to }{\mathrm{SO}}_{4}^{-}\cdot +{\mathrm{SO}}_{4}^{2-}$$

One of the most important factors for a successful ISCO field application is the effective delivery of oxidants, such as PS, to contaminated sites. Conventional PS delivery approaches include well injection, pressurized direct-push technology and recirculation systems. However, injecting PS as an aqueous solution can be challenging due to non-selective consumption and the uneven distribution caused by preferential pathways in the subsurface [17]. To address this issue, slow-release materials can be utilized [18]. The combination of ISCO and slow-release technology can prolong the life of oxidants, boosting the oxidation efficiency and thus reducing the cost of excessive usage of oxidants and secondary pollution. For this reason, researchers are focusing on the potential of sustained-release materials, including persulfate columnar candles [11], potassium permanganate (KMnO_4_)-releasing composites (PCRs) [19], new controlled-release permanganate particles [20] and persulfate-activator soft solids [21], which could continuously release oxidants to contaminated sites and remove organic pollutants. Most sustained-release materials are composed of oxidant agents and a matrix [22]. Paraffin wax is non-toxic and easy to mold into different shapes. It was initially employed to formulate matrix fertilizer and is now commonly employed as a matrix to prepare sustained-release materials [23,24]. Christenson et al. (2016) developed sustained-release permanganate candles and conducted a five-year field experiment, achieving a reduction of 89% in the TCE concentration [25]. Rauscher et al. (2012) used permanganate candles to remediate polycyclic aromatic hydrocarbon (PAH)-contaminated water; the results showed that the majority of the 16 PAHs tested were degraded within 2–4 h [26]. Wang et al. (2009) selected ethyl cellulose and paraffin wax as wall materials to prepare an encapsulated potassium ferrate, and the 2-sec-butyl-4,6-dinitrophenol degradation efficiency was 93% at the optimal microcapsule concentration and pH for 80 min [27]. In addition, paraffin wax is environmentally benign and can be completely degraded in soil in several weeks [28,29]. Subsequently, persulfate sustained-release material containing persulfate as an oxidant and paraffin wax as a matrix has recently been the focus of research due to its effectiveness in removing organic pollutants from groundwater [30,31]. However, there are fewer studies on the dynamic release characteristics and release kinetics of persulfate sustained-release materials in a laboratory setting. Columnar sustained-release material has a smaller specific surface area and a longer release time compared to other shapes, such as capsules, pellets and gels. In this research, a persulfate sustained-release candle with paraffin wax as the matrix (PS@PW) was synthesized using the melt injection molding condensation method, and the PS dynamic release characteristics and kinetics were investigated through column experiments. The matrix boundary diffusion model was employed to fit the PS diffusion coefficient from PS@PW, and the release longevity of the PS@PW was then predicted.

## 2. Materials and Methods

### 2.1. Chemicals

All chemical reagents used in the experiment were of analytical grade. Sodium persulfate (Na_2_S_2_O_8_, ≥99.0%) was purchased from Macklin Biochemical Co., Ltd. (Shanghai, China). Paraffin wax (white to off-white solid; melting point, 48–50 °C; flash point, 113 °C) was purchased from Aladdin BioChem Technology (Shanghai, China). Potassium iodide (KI, ≥99.0%) and sodium bicarbonate (NaHCO_3_ ≥ 99.5%) were purchased from Xilong Scientific Co., Ltd. (Shantou, China). All solutions were prepared using deionized water.

### 2.2. PS@PW Preparation

PS@PW was prepared using the melt injection molding condensation method, an improved method that has been previously reported [3,32]. The procedure is as follows: Sodium persulfate (Na_2_S_2_O_8_) was ground and sieved through a 100-mesh sieve. The paraffin wax was chopped into a beaker and heated to 70 °C until it melted. Then, ground Na_2_S_2_O_8_ powder was added to the melted wax, and the mixture was stirred well and quickly poured into a columnar Plexiglas mold that had been sprayed with a demolding agent in advance. Finally, the mixture was cooled to room temperature and extracted from the mold to obtain PS@PW with a PS/PW mass ratio of 2:1.

### 2.3. Dynamic Column Experiments

#### 2.3.1. Experimental Procedure

The main part of the experimental device was six identical glass columns (500 mm in length, 30 mm in inner diameter), which were filled with glass beads (1 mm in diameter) to ensure uniformity of water distribution throughout the apparatus. A schematic of the setup for the column experiment is presented in Figure 1. Each glass column had a PS@PW sample of the same length but with different diameters, and the PS@PW was placed in the center of the glass columns, surrounded by glass beads from top to bottom. To prevent the loss of glass beads from the outlets, thick permeable material such as gauze was placed in front of the outlets. After PS@PW and the glass beads were filled, tinfoil was employed to completely cover the glass columns, outlet pipes and reservoirs to prevent the effects of light. Deionized water was pumped into the columns from top to bottom using a peristaltic pump at a steady flow rate of 2 mL/min, and the effluent was collected in the reservoirs. Samples were collected at predetermined times from the outlets, and the concentration of PS was determined via UV spectrophotometry, according to the spectrophotometric method proposed by Liang et al. [33]. The solution volume of reservoirs was measured with a measuring tube, and the pH was measured with a pH meter.

#### 2.3.2. Parameters of PS@PW

The release of PS from PS@PW with varying diameters of 10 mm, 14 mm and 22 mm was investigated through experiments while maintaining a PS/PW mass ratio of 2:1, as our previous research demonstrated that PS@PW with a PS/PW mass ratio of 2:1 has a better release performance [34]. The detailed parameters of the PS@PW in the six columns are presented in Table 1.

### 2.4. Dynamic Release Kinetics Fitting Model

First-order and second-order kinetic models were selected to fit the PS@PW dynamic release, as shown in Equations (3) and (4) [6,35].
(3)$$\mathrm{Ln}\left(1-\frac{q}{q_{e}}\right)=-k_{1}t$$
(4)$$\frac{t}{q}=\frac{1}{k_{2}q_{e}^{2}}+\frac{t}{q_{e}}$$
where

*q_e_* is the maximum release mass of PS (g);

*q* is the release mass of PS at time t (g);

t is release time (d);

*k*_1_ is the first-order kinetic release rate constant (d^−1^);

*k*_2_ is the second-order kinetic release rate constant (g^−1^d^−1^).

### 2.5. Establish Matrix Boundary Diffusion Model

The columnar PS@PW is a mixture of persulfate and paraffin wax, with a PS concentration that surpasses the PS solubility threshold in water. In a flowing-water system, the columnar PS@PW continuously transports PS to the water, while the material retains its original shape. In addition, dissolution and diffusion are the main drivers of PS release from the material. Therefore, assuming that PS is homogeneously distributed in the candle with a constant diffusion coefficient, the kinetics of PS release can be quantified using a matrix-boundary diffusion model developed by Roseman and Higuchi [36], as shown in Figure 2, similar to the previous study by Liang et al. (2017) [30].

In Figure 2, the shaded area represents the saturated area of PS, and the blank area represents the fully released area of PS. The initial release of PS@PW is completely PS-saturated with a saturation zone radius of *r*_0_ (i.e., *r* = *r*_0_). As PS is gradually released from the PS@PW, the radius of the saturation zone I diminishes, eventually reaching zero (i.e., *r* = 0) when all the PS has been released. Assuming that the PS in the fluid moves rapidly due to advection, dispersion and reaction, it can be concluded that the concentration of the PS is zero (i.e., *C_b_* = 0) at *r* = *r_a_*.

According to Fick’s law, the rate of PS passing through the side surface area of the columnar PS@PW can be determined using Equation (5) [30].
(5)$$\frac{dM}{dt}=2\pi rhD_{e}\frac{dC}{dr}$$
where

*M* is the cumulative release mass of PS from the PS@PW (g);

*t* is the duration of the continuous release of PS from the PS@PW (d);

*h* is the height of the PS@PW (cm);

*r* is the radius of the saturated zone in the PS@PW (cm);

*D_e_* is the PS effective diffusion coefficient in the PS@PW (cm^2^/d);

*C* is the concentration of PS in the aqueous phase (g/cm^3^).

When Equation (5) satisfies the boundary conditions *C*_0_ = *C_s_* and *r* = *r*_0_, Equations (6) and (7) can be derived from Figure 2 and Equation (5):(6)$$\frac{D_{e}C_{s}t}{A}=\frac{r^{2}}{2}ln\frac{r}{r_{0}}+\frac{1}{4}\left(r_{0}^{2}-r^{2}\right)$$
(7)$$\;M=\pi hA\left(r_{0}^{2}-r^{2}\right)$$
where

*C*_0_ is the concentration of PS at the material boundary;

*C_s_* is the solubility of PS in water at 20 °C (556 g/L);

*r*_0_ is the initial radius of the candle (cm);

*A* is the mass of PS in the unit volume of the PS@PW (g/cm^3^).

## 3. Results and Discussion

### 3.1. Dynamic Column Experiment of Different Sizes

The dynamic column experiment was conducted for a maximum of 21 days, during which the release rate of PS from PS@PW was monitored. As depicted in Figure 3, the maximum release time for PS@PW with diameters of 10 mm, 14 mm and 22 mm was 14 days, 21 days and 21 days, and the time needed to release 50% of PS was 2 days, 4 days and 6 days, respectively. PS was not detected in the effluent, indicating the cessation of PS release. It is evident that the diameter of PS@PW played a pivotal role in determining the PS release rate and amount, as a smaller diameter facilitates a faster and more thorough release.

The release rate (g/d) and specific release rate (g/d/g) are shown in Figure 4a,b. The average release rates of the three diameters (10 mm, 14 mm and 22 mm) of materials on the first day were 1.70 g/d, 2.54 g/d and 3.94 g/d, respectively. From the second day to the time of the maximum release ratio, the average release rates decreased to 0.377 g/d, 0.611 g/d and 1.55 g/d. The results suggest that the release rate of PS@PW exhibited an initial rise, followed by a subsequent decline in the medium term, ultimately reaching a state of complete cessation. Moreover, no significant difference was seen in specific release rates for PS@PW with any of the three diameters, suggesting a clear correlation between larger diameters and faster rates of release.

The PS concentrations in the effluent are shown in Figure 4c. After 90 min of operation, the glass columns were completely filled and the outlets began to discharge water, and the PS concentration in the effluent was 0.02 g/L, 1.93 g/L and 0.06 g/L for PS@PW with diameters of 10 mm, 14 mm and 22 mm, respectively. The maximum PS concentration in the outlets for the 10 mm, 14 mm and 22 mm diameters during the experiment was 2.36 g/L, 1.98 g/L and 6.18 g/L, respectively. Subsequently, the concentration of PS in the effluent decreased and eventually stabilized. It can be observed that the PS concentration in outlets of different-diameter materials followed a similar pattern: increasing and then decreasing in the early stage of release (first day of release), and thereafter tending to remain stable for a prolonged period. Beyond that, when comparing the concentration of PS in the outlets of different-diameter materials at the same time, it is obvious that the larger the diameter, the higher the concentration of PS in the outlet.

The variation in PS concentration in the effluent can be explained by the dissolution–diffusion process [4,37]. At the initial stage of release, the PS concentration in the effluent was lower as there was a lack of adequate water surrounding PS@PW, leading to a smaller contact area and less time before the PS@PW and flowing water came into contact. The amount of water in the glass column increased with time until PS@PW was fully immersed, resulting in an increase in the contact area between the PS@PW and water, allowing for a more prolonged interaction. Consequently, PS was released more easily into the water, and the concentration of PS increased. At the mid- to late stage of the release process, the PS on the surface of the material was completely released, resulting in the formation of pores. Simultaneously, the inner PS dissolved when it came into contact with water, which caused the release rate of PS from the material to slow down and become stable. As a result, the PS concentration in the outlet remained constant.

The pH of the effluent in the reservoirs depicted in Figure 4d showed a pattern of fluctuation. The pH began to rise, followed by a period of stability, and then another upsurge, before attaining a steady state. It is speculated that this variation was due to the concentration of PS at the outlets; initially, the outlets had a low pH because of the high PS concentration; however, as time passed, the PS concentration decreased, thus leading to an increase in pH.

### 3.2. Dynamic Release Kinetics Fitting Results

The first-order and second-order kinetics were used to fit the PS release process from PS@PW with three different diameters, as presented in Table 2 and Figure 5. The second-order kinetic equation exhibited a higher R-squared (*R*^2^) than the first-order one, indicating that the release of PS under dynamic conditions followed the second-order kinetic equation. As PS@PW underwent a dissolution–diffusion process, it produced numerous micro-pores, which enabled water to enter and dissolve the PS@PW core. This led to a quick initial release of PS followed by a slower release due to the diffusion of water into the PS@PW [4]. The process of dissolution and diffusion is well described by second-order kinetics [38]. In addition, the second-order kinetic fitting diffusion rate constants (*k*_2_) were inversely proportional to the diameters, demonstrating that the diameter of the PS@PW significantly influenced its release performance.

### 3.3. Matrix Boundary Diffusion Model Fitting

The dynamic release results for PS@PW (D = 22 mm) and Equations (6) and (7) were employed to fit the experimental results with varying effective diffusion coefficients (*De*). To achieve a good fit, it is imperative that the following criteria be adhered to: (1) the discrepancy between the simulated and measured mass release over time must be minimized and (2) the difference between the simulated and measured release longevity of the PS@PW must also be minimized [39].

For the fitting process, 0.02 cm^2^/d, 0.024 cm^2^/d, 0.028 cm^2^/d and 0.032 cm^2^/d were selected as the effective diffusion coefficients (*De*). Combining Equations (6) and (7) and the *De*, the function relationship between the cumulative release mass and release time of the PS@PW was identified, and then the maximum sustained release times (i.e., release longevity) of PS@PW with different *De* values were obtained, as shown in Figure 6 and Table 3.

The release longevity of PS@PW (D = 22 mm) was 21 days, according to the column experiment results, and the results for the fitting of four distinct diffusion coefficients were 30 days, 26 days, 22 days and 20 days. It was determined that a greater diffusion coefficient leads to a shorter lifespan of PS@PW. However, during later stages, the rate of release was too low, and the amount of PS released during this period was negligible. To better quantify the release performance, the time taken for PS@PW to reach half release (*t*_1/2_), 95% release (*t*_95%_) and full release (*t*_100%_) was calculated. The *De* of the PS@PW with a diameter of 22 mm was estimated to be between 0.028 cm^2^/d and 0.032 cm^2^/d.

According to Figure 6b, the estimated release lifespan for a range of *De* values from 0.01 cm^2^/d to 0.04 cm^2^/d spanned from 61 to 16 days, and a *De* value of approximately 0.028 cm^2^/d corresponded to a release period of 21 days.

### 3.4. PS@PW Application Scene

#### 3.4.1. Field Application Methods

Persulfate sustained-release materials are mainly employed for ISCO of groundwater and are commonly used for the permeable reaction barrier [40]. A permeable reaction barrier can be constructed by conducting a pollutant and hydrogeological survey of the contaminated sites to determine the number and spacing of construction boreholes. The boreholes should then be drilled, and a layer of screen should be installed around the borehole walls to maintain the shape of the boreholes while allowing adequate water to pass through. Finally, PS@PW should be inserted into the boreholes and secured to the appropriate depth. Once the release of PS@PW is complete, it should be removed and replaced with fresh PS@PW.

#### 3.4.2. Release Longevity Expectancy

The release longevity of PS@PW is critical to the remediation of contaminated sites. When PS@PW reaches the maximum release time (i.e., *r* = 0), based on the *D_e_* of the PS@PW derived from the model fitting (*D_e_* = 0.028 cm^2^/d, *W*_PS_:*W*_PW_ = 2:1), Equation (6) can be used to obtain Equations (8) and (9).
(8)$$\frac{D_{e}C_{s}t}{A}=\frac{1}{4}r_{0}^{2}$$
(9)$$t=\frac{r_{0}^{2}A}{4D_{e}C_{s}}$$
where

*D_e_* is the PS effective diffusion coefficient in the PS@PW (cm^2^/d);

*C_s_* is the solubility of PS in water at 20 °C (556 g/L);

*t* is the duration of the continuous PS release from the PS@PW (d).

Assuming the groundwater conditions are analogous to those of the column experiment, that is, *W*_PS_:*W*_PW_ = 2:1, and the PS@PW is 500 mm long, the release longevities of materials with diameters of 80 mm, 100 mm and 120 mm were predicted based on the model, as shown in Table 4. According to Equation (6), the cumulative release mass of the three materials varied over time, as shown in Figure 7.

## 4. Conclusions

To achieve a successful implementation of sustained-release material (PS@PW) technology, it is essential to have a comprehensive understanding of the kinetics and mechanisms governing the sustained release of reactive agents. In this research, the melt injection molding condensation method was used to prepare PS@PW samples with three diameters (10 mm, 14 mm and 22 mm). The dynamic release column experiment was employed to assess the dynamic release characteristics and kinetics of PS@PW. The following conclusions were reached:

(1) A smaller diameter of PS@PW enabled a faster and more thorough release than the larger diameter, which led to a longer release longevity;

(2) The release process of PS followed the second-order kinetics, and the release rate constant was positively correlated with the diameter of PS@PW;

(3) The matrix boundary diffusion model was utilized for determining the effective diffusion coefficient of PS from PS@PW, which was found to be 0.028 cm^2^/d for the PS@PW with a 22 mm diameter. The release longevity of PS@PW with a length of 500 mm and a diameter of more than 80 mm was predicted to be more than 280 days.

Based on our results, a well-designed PS@PW has the ability to consistently release PS for a prolonged period of time, thus representing a high-efficiency and long-term groundwater remediation, and has important guiding significance for in situ chemical oxidation.

## Figures and Tables

**Figure 1 toxics-11-00829-f001:**
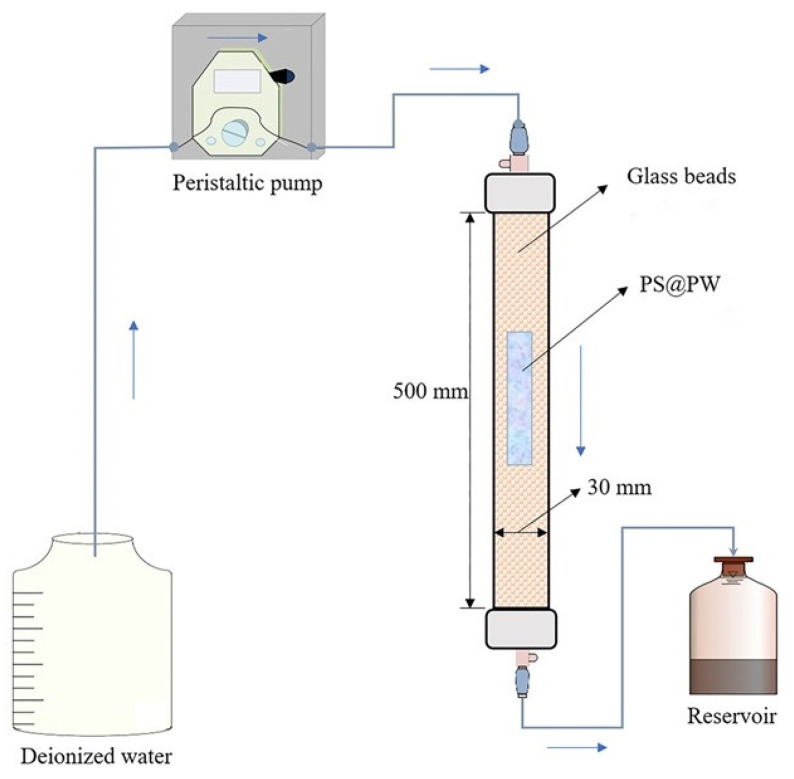
Schematic of the setup for the column experiment.

**Figure 2 toxics-11-00829-f002:**
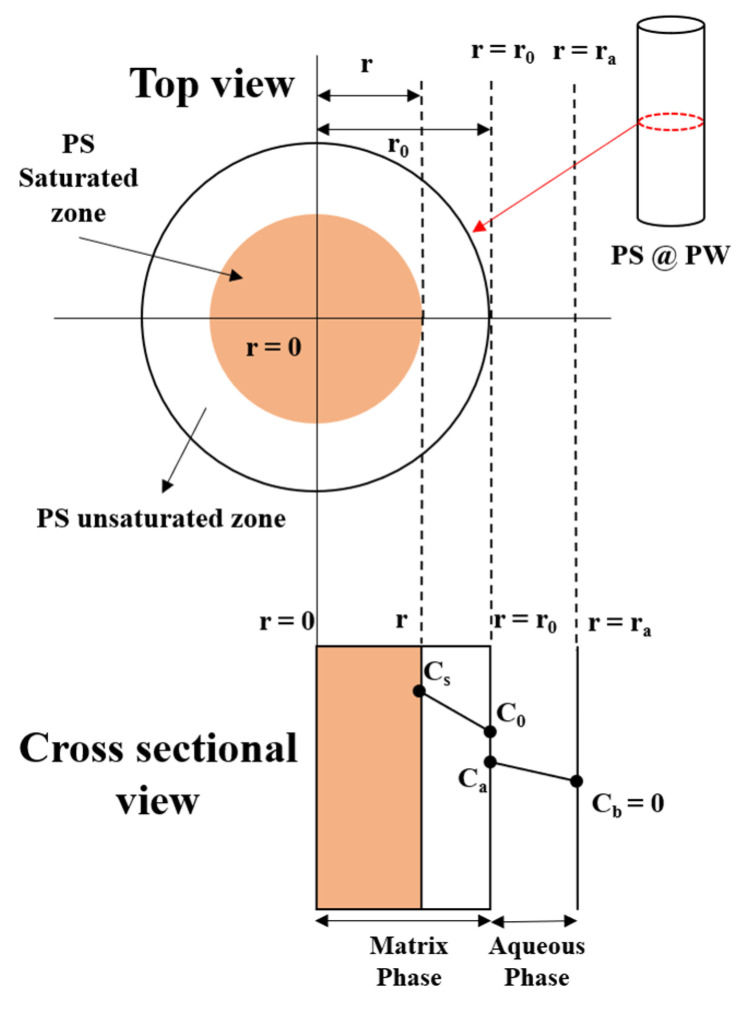
Matrix-boundary diffusion model concept diagram of PS@PW release.

**Figure 3 toxics-11-00829-f003:**
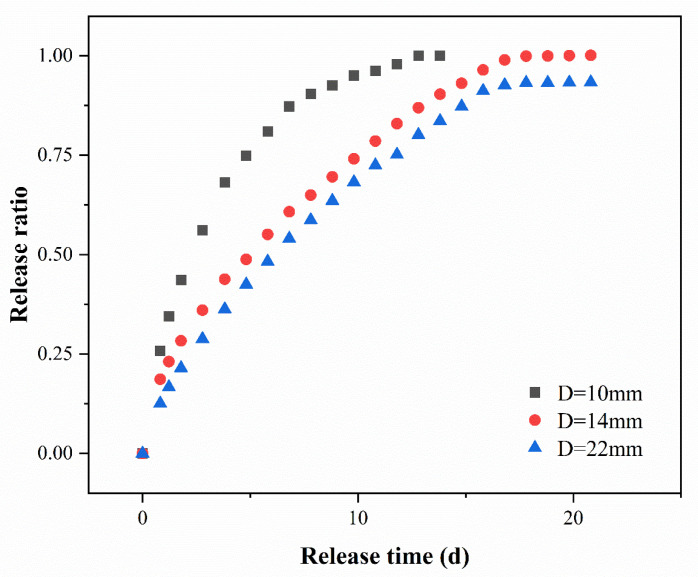
Dynamic release ratio of PS@PW with different diameters.

**Figure 4 toxics-11-00829-f004:**
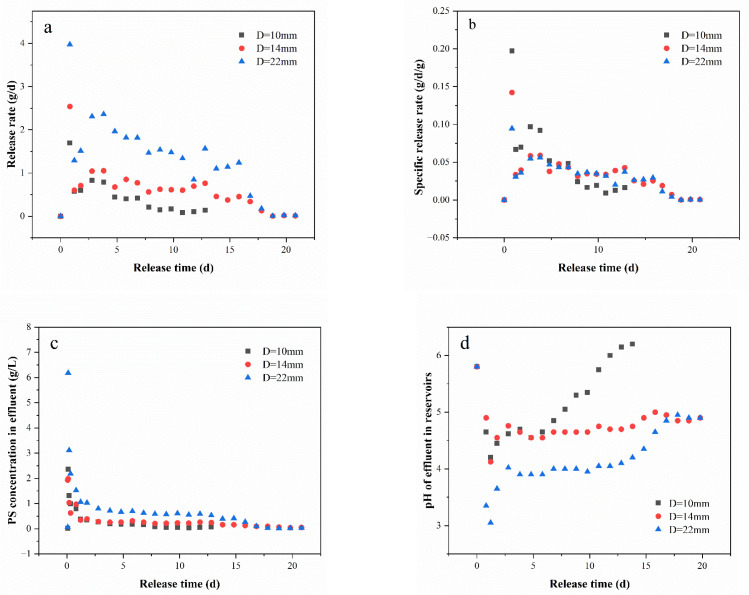
(**a**) Release rates of PS@PW with three diameters, (**b**) specific release rates of PS@PW with three diameters, (**c**) PS concentration in the outlets and (**d**) pH of effluent in reservoirs.

**Figure 5 toxics-11-00829-f005:**
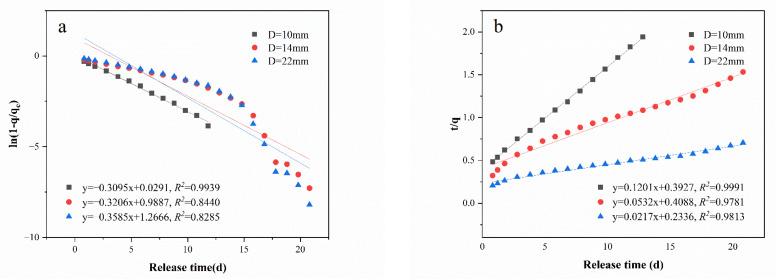
(**a**) PS@PW dynamic release first-order equation fitting and (**b**) PS@PW dynamic release second-order equation fitting.

**Figure 6 toxics-11-00829-f006:**
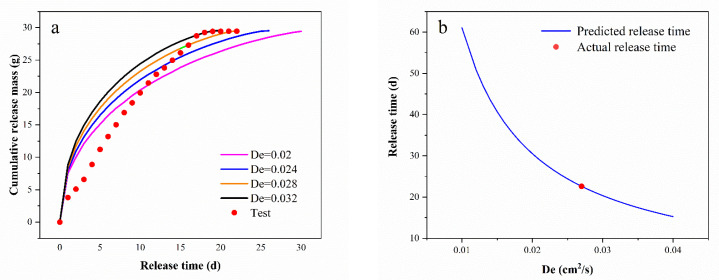
(**a**) PS cumulative release mass and measured value fitted by different *De* values and (**b**) predicted release longevity of PS@PW with different *De* values.

**Figure 7 toxics-11-00829-f007:**
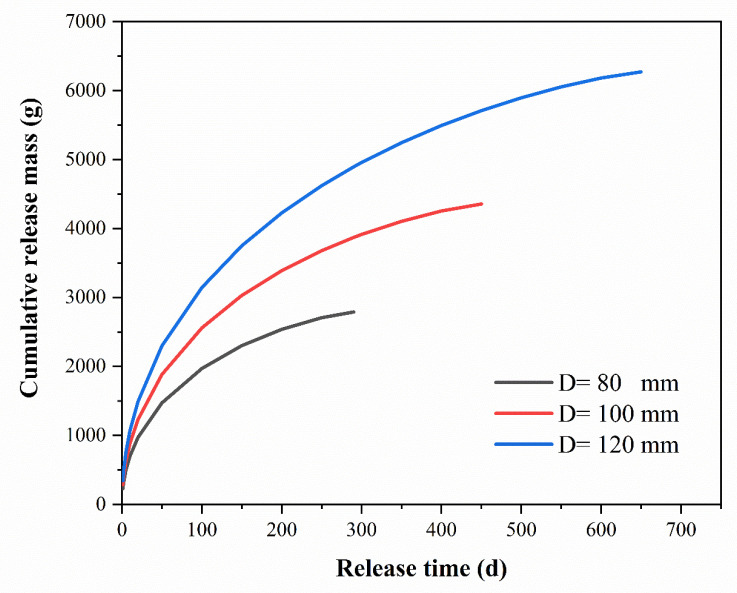
Prediction of cumulative released PS quality.

**Table 1 toxics-11-00829-t001:** Parameters of PS@PW in the six columns.

Serial Number	High (mm)	Diameter (mm)	Weight (g)	PS Content (g)
1	70	10	8.55	5.70
2	70	10	8.64	5.76
3	70	14	17.90	11.93
4	70	14	17.80	11.87
5	70	22	41.90	27.93
6	70	22	42.20	28.13

**Table 2 toxics-11-00829-t002:** PS@PW column experimental release kinetic reference values.

Diameter	First-Order Kinetic Fitting		Second-Order Kinetic Fitting	
	*k*_1_ (d^−1^)	*R* ^2^	*k*_2_ (g^−1^d^−1^)	*R* ^2^
D = 10 mm	−0.3095	0.9939	0.1201	0.9991
D = 14 mm	−0.3206	0.8440	0.0532	0.9781
D = 22 mm	−0.3585	0.8285	0.0217	0.9813

**Table 3 toxics-11-00829-t003:** Fitting key parameter values.

Group	*De* (cm^2^/d)	*t*_1/2_ (d)	*t*_95%_(d)	*t*_100%_(d)
1	0.020	4.8	24.5	30
2	0.024	3.7	20.5	26
3	0.028	3.5	17.5	22
4	0.032	2.8	15.5	20
5	Test	5.3	16.7	21

**Table 4 toxics-11-00829-t004:** Release longevity of materials with diameters of 80 mm, 100 mm and 120 mm.

Material Diameter (mm)	80	100	120
Predicted release longevity (d)	288	450	648

## Data Availability

The raw/processed data required to reproduce the above findings can not be shared at this time as the data also form part of an ongoing study.

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
