# Peer review of "Dynamic Release Characteristics and Kinetics of a Persulfate Sustained-Release Material"

_toxics, 2023, doi:10.3390/toxics11100829_

Round 1
Reviewer 1 Report
Zhu et al described Dynamic release characteristics and kinetics of a persulfate sustained-release material. The manuscript written considerably okay. I would recommend few modifications to further improve the quality of the manuscript, as below.
1. Chapter 2.2 : please add some references for PS@PW preparation techniques that the author followed.
2. Figure 1 please revise the diameter size : 10 mm, 14mm and 22mm
3. Table 1. Please explain about ratio PS:PW in the paragraph
4. In the equation 3 and 4 there are ‘t’ as a factor in the equation, please give information of ‘t’
5. Line 186, 0.02, 1.93 and 0.06 these value belongs which diameter? Similarly with line 187.
6. Please add more references for line 195 to 207
7. Table 2, why unit k2 for second order kinetic fitting different with equation 4
8. Figure 5, the graph very small, please enlarge in order to see the information inside the graph clearly.
9. Please add references for line230 to 235
10. What is D_e C_s t in equation 8 and 9, please explain.
11. A lot of typo and formula errors ( Ex. E0 -à Eo, SO4-. -à SO4-. Etc
12. It is highly recommended to include laboratory set up images used for this study.
13. What efforts are already made earlier in the literature to slow release of persulfate. Compare your study with other literature.
14. The values inside the figure 5 are unreadable.
A lot of typo errors
Reviewer 2 Report
The paper entitled “Dynamic release characteristics and kinetics of a persulfate sustained-release material” reported preparation of a cylindrical sustained-release material was prepared with paraffin wax and sodium persulfate. The authors prove based on experimental results, that a well-designed PS@PW has the ability to consistently release reagent for a prolonged period of time, thus representing a high-efficiency and long-term groundwater remediation technology, and has an important guiding significance for ISCO. The experimental section and conclusions were well described, a lot of experimental tests explained scientifically, and thus in summary this manuscript is publishable in Toxics Journal
1. The authors should better evidenced in the abstract section the novelty of this study? 2. In the introduction section, the references are not according to the "Instructions for authors" requested by the journal. In the text, reference numbers should be placed in square brackets, please correct. 3. Complete the introduction with more previous studies about the using of PS into the degradation processes. The authors need to use more references. 4. All used equations should have a reference. Please complete the manuscript. 5.The insets from Fig 5 are not clear. Please put them separately for a better understanding 6. References need to be in journal format.
In summary, this manuscript is publishable on Toxics Journal if some minor issues have been corrected.
Reviewer 3 Report
The manuscript entitled “Dynamic release characteristics and kinetics of a persulfate sustained-release material” concerns the preparation of cylindrical sustained-release material from paraffin wax and sodium persulfate. In addition, the Authors conducted the dynamic release column experiment to study the effect of the diameter of the resulting material and sodium persulfate mass ratio on sodium persulfate release. In my opinion, the research presented in the manuscript could be of interest to readers of the Toxics.
The manuscript is clearly written. The results are presented in a comprehensible way. The tables and figures present essential data. However, there are several points that must be considered to improve the quality of the manuscript.
- Why was the ratio of Na2S2O8 to paraffin wax set at 2? Why were other variables not checked, such as the particle size of the persulfate, the ratio of Na2S2O8 to paraffin wax, the composition of the paraffin wax? Please note that there are many types of paraffin waxes, so detailed characteristics of such material (composition, viscosity, etc.) should be provided.
- What do the error bars (Figures 2 and 3) represent? Is it the standard deviation between two repeated experiments, or something else (such as measurement uncertainty)?
- The Authors wrote in lines 218-221: “The second-order kinetic equation exhibited a higher R2 than the first-order, which indicated that the release of PS under dynamic conditions followed the second-order kinetic equation.” Please note that the relationships shown in Figure 5b for D=14 and 22 mm are not linear, so the kinetics of this process must be more complex. Please comment on this observation and try to explain this behavior.
- Is a sustained-release persulfate candle with paraffin wax as a matrix used in practice, or is this just laboratory research to develop a slow-release method for persulfate? Please add information about the risk of releasing paraffin wax into the environment and the possible effects of such a release?
- Persulfate is a strong oxidizer, so the stability of such a candle should be considered. Is it safe to store such a candle over time? Is persulfate stable during storage of such a candle, or can it react with paraffin wax? Did the Authors check the stability of such a candle during storage?
- Line 41: It should be "Equations (1) and (2)" instead of "Equtions(1) and (2)".
- Line 105: It should be "pH meter" instead of “PH meter”.
- Please use subscripts when writing chemical formulas (e.g., lines 40, 42, 76) and parameters designations (e.g., lines 219, 221).
- The data contained in the tables in Fig. 5a and 5b are hardly visible. Please, enlarge the figures or make separate tables.
- Lines 236, 249, 251, 273 and 307: Please use superscript when writing cm2.
Overall, the manuscript presents quite interesting results and is worth to consider for publication in the Toxics.
Reviewer 4 Report
This paper presents a kinetic analysis of PS release using PS@PW and shows that PS@PW can be used for a long period of time through simulation; it provides useful insights for the remediation of organic contaminants in groundwater using PS.
1) Line 59: SPS and SR are represented in the text without definitions. Please define them before use.
2) Line 67-72: paraffin wax is used as a sustained-release material, please describe the advantages of using PW and why the authors chose it.
3) Table 2: Why is this release reaction a second-order reaction and not a first-order reaction? Explain why.
4) Fig. 5: The table in Fig. 5 is too small to read.
5) Please define the abbreviations used at the end of the sentence.
Round 2
Reviewer 3 Report
I believe that the authors have appropriately addressed my comments and revised the manuscript accordingly. The revised manuscript can be accepted for publication.